# Effect of Process Parameters on Void Distribution, Volume Fraction, and Sphericity within the Bead Microstructure of Large-Area Additive Manufacturing Polymer Composites

**DOI:** 10.3390/polym14235107

**Published:** 2022-11-24

**Authors:** Neshat Sayah, Douglas E. Smith

**Affiliations:** Department of Mechanical Engineering, Baylor University, Waco, TX 76706, USA

**Keywords:** additive manufacturing, carbon-fiber-reinforced polymer composites, micro-computed tomography (µCT), microstructural voids, void sphericity

## Abstract

Short carbon fiber-reinforced composite materials produced by large-area additive manufacturing (LAAM) are attractive due to their lightweight, favorable mechanical properties, multifunctional applications, and low manufacturing costs. However, the physical and mechanical properties of short carbon-fiber-reinforced composites 3D printed via LAAM systems remain below expectations due in part to the void formation within the bead microstructure. This study aimed to assess void characteristics including volume fraction and sphericity within the microstructure of 13 wt% short carbon fiber acrylonitrile butadiene styrene (SCF/ABS). Our study evaluated SCF/ABS as a pellet, a single freely extruded strand, a regularly deposited single bead, and a single bead manufactured with a roller during the printing process using a high-resolution 3D micro-computed tomography (µCT) system. Micro voids were shown to exist within the microstructure of the SCF/ABS pellet and tended to become more prevalent in a single freely extruded strand which showed the highest void volume fraction among all the samples studied. Results also showed that deposition on the print bed reduced the void volume fraction and applying a roller during the printing process caused a further reduction in the void volume fraction. This study also reports the void’s shape within the microstructure in terms of sphericity which indicated that SCF/ABS single freely extruded strands had the highest mean void sphericity (voids tend to be more spherical). Moreover, this study evaluated the effect of printing process parameters, including nozzle temperature, extrusion speed and nozzle height above the printing table on the void volume fraction and sphericity within the microstructure of regularly deposited single beads.

## 1. Introduction

Additive manufacturing (AM), also known as the three-dimensional (3D) printing, is a process of adding materials layer upon layer to fabricate objects from 3D digital models. AM technology has attracted considerable attention in recent years due to its potential to incorporate design detail with little or no additional manufacturing costs and reducing material waste [1]. There are several types of additive manufacturing processes, among which polymer extrusion/deposition, which includes fused filament fabrication (FFF), is one of the most popular AM methods due to the availability of a wide range of neat thermoplastics and thermoplastic composite materials [2,3]. The large-area additive manufacturing (LAAM) system is a polymer extrusion/deposition process that utilizes a pellet fed polymer extruder system providing high flow rates during the deposition process and a wide selection of low-cost materials. LAAM systems melt thermoplastic pellets which are then conveyed through a single screw extruder and then deposited on a build plate or platform, where it cools and solidifies [4].

One advantage of LAAM 3D printing is the availability of polymers and polymer composites. Acrylonitrile butadiene styrene (ABS) is a thermoplastic polymer that is commonly 3D printed due to its high rigidity, high dimensional stability, and favorable electrical insulating properties [5,6]. Apart from these properties, AM products produced with ABS are limited due to reduced mechanical properties and excessive thermal distortion [7,8]. The addition of short carbon fibers (SCF) into the polymer matrix improves the stiffness and toughness of the final additively manufactured part making SCF/ABS highly desirable in LAAM applications. Moreover, carbon fiber inclusions significantly reduce the coefficient of thermal expansion and increase the thermal conductivity of the deposited composite bead which improves dimensional stability and decreases warping of the 3D-printed parts. As a result, SCF/ABS composites in polymer extrusion/deposition systems not only improves the mechanical and thermal performance of the final parts but also reduces the manufacturing time, and cost of production [9]. Unfortunately, it has been shown that the mechanical properties of the carbon-fiber-reinforced (SCF/ABS) composite remain below expectations due to inferior interlayer adhesion, the formation of voids between beads, and uncontrolled fiber orientation and micro voids within the final AM part’s bead microstructure [9,10]. The addition of SCF changes the polymer rheology and increases the viscosity of the polymer composite melt, which tends to result in an increased micro void volume fraction in the AM composite [6,11]. Micro voids are closed volumes within the SCF polymer composite microstructure that are not filled with polymer matrix and fibers [11,12]. The microstructural comparison of the neat polymer and carbon-fiber-reinforced polymer composite shows that the micro void content is quite low and/or negligible within the neat polymers [7]. However, with the addition of carbon fibers into the polymer, micro voids increase substantially due in part to differences in the coefficient of thermal expansion between the fiber and matrix [7,13]. The formation of voids during the polymer extrusion/deposition printing process reduces the density and strength of the composite material, in addition to creating points of stress concentration, which results in lower mechanical properties in the final AM part [7,14].

Previous work by Yeole et al. [15] compared the performance of the additively manufactured CF/PPS component to that of traditional processing methods, including injection molding (IM) and extrusion–compression modeling (ECM). Results indicated that AM components contain a significantly higher number of micro voids than parts produced using these traditional processing methods. Denault et al. [16] also reported the lower tensile properties of the CF/ABS-manufactured part using FFF than the compression molded part due to the higher amount of micro voids (porosity) within the microstructure. Tekinalp et al. [7] compared the void content within the microstructure of the compression molded (CM) and fused deposition modeling (FDM) CF/ABS, in which the CM part indicated no visible void content. In contrast, the FDM part exhibited a significant number of pores within the bead microstructure.

Various techniques, including destructive and non-destructive testing techniques, have been employed to study voids in composite materials. Density determination is typically a destructive technique that gives a void content value in the form of a void volume fraction. Based on ASTM D2734, the void volume fraction can be calculated from the relative difference between theoretical and measured composite density as
(1)$$Vv=100-\rho_{c}^{m}\left(\frac{W_{r}}{\rho_{r}}+\frac{W_{f}}{\rho_{f}}\right)$$
where Vv, $\rho_{c}^{m}$, *W*, ρ, r and f are the void volume fraction, measured composite density, weight percentage, density, resin, and fiber, respectively. Unfortunately, this technique only provides information on the sample’s average void volume fraction and cannot quantify void size, shape, and spatial distribution within the composite microstructure [17]. Scanning electron microscope (SEM) techniques are also used to determine void content in composite materials. In contrast to the density measurement technique, SEM measurements allow for the characterization of the shape, size, and spatial distribution of voids. Unfortunately, SEM measurements are dependent on the direction of the two-dimensional (2D) slice chosen for the analysis, yielding results that are view-dependent [6,11]. Alternatively, micro-computed tomography (µCT) has emerged as a promising non-destructive technique for 3D visualization of the matrix, fibers, and voids, including their size, shape, orientation, and spatial distribution within the composite microstructure. µCT provides three-dimensional (3D) information about a material sample by combining X-ray images from multiple object angles. Unlike SEM, µCT is not affected by section orientation. However, the accuracy of µCT results is highly dependent on the image processing technique [11,18,19]. Other researchers, including Diouf-Lewis et al. [20], Dana et al. [21], and Yang et al. [6], have reported the void volume fraction within the microstructure of AM polymer composites obtained via µCT analysis. However, there is still a lack of information on the directional dependence of void distribution and void shape within the microstructure.

This paper presents an experimental study on void volume fraction, void shape, and directional dependence of void distribution within the microstructure of SCF/ABS samples manufactured using LAAM polymer composite extrusion/deposition AM. The microstructure of SCF/ABS samples including a pellet, a single freely extruded strand, a regularly deposited single bead, and a deposited single bead manufactured using a roller during the printing process were evaluated through µCT inspection techniques where image analysis was performed to evaluate void features within the polymer composite microstructure. Moreover, this paper considers the effect of printing process parameters including, nozzle temperature, extrusion speed, and nozzle height above the print bed on the void volume fraction within the microstructure of SCF/ABS.

## 2. Materials and Methods

### 2.1. 3D Printing Process of SCF/ABS

The custom-built LAAM system shown in Figure 1 was used to print short fiber polymer composite bead samples required for this study. The extruder used in our LAAM system was a Strangpresse Model 19 (Strangpresse, Youngstown, OH, USA) [22], where the print path was defined from G-code with Mach3 software (Newfangled Solutions LLC, Livermore Fls, ME, USA). The LAAM print volume was 48” × 48” × 6” [23]. PolyOne (Avient, Avon Lake, OH, USA) SCF/ABS composite was used in this study where the carbon fiber diameter was seven microns, and the carbon fiber weight percent was 13 (wt%). Carbon fiber-filled ABS pellets were dried in a convection oven at 80–85 °C for eight hours and then placed in a dry ambient chamber at room prior to the LAAM 3D printing process. Beads of material were extruded and deposited onto a tape-covered aluminum print surface using the Strangpresse Model 19 extruder which had three temperature zones along the length and operated at 90 revolutions per minute (rpm). Various 13 (wt%) SCF/ABS sample types were prepared for micro void analysis which included: (1) a pellet of the material as received from PolyOne, (2) a single strand freely extruded in air without deposition, (3) a single bead deposited onto the print surface, and (4) a single bead extruded onto the print surface and then compressed with a roller immediately following the extrusion/deposition nozzle. For the freely extruded single strand, the nozzle height (the distance between the nozzle tip and the print bed) was set to 6 inches where a test sample was extracted during free extrusion so as not to stretch the strand. To fabricate our regular bead and roller-compressed bead samples, the nozzle height of the extruder was set at 1.2 mm above the print table with a nozzle translation speed with respect to the print surface of 240 cm/min. To prepare roller-compressed beads, a plastic roller with the diameter of 3 cm was placed 4 cm behind the nozzle tip to partially compress the printed bead onto the print bed (cf. Figure 1b). The roller was sufficiently wide to cover the entire width of the bead. The roller height (i.e., the minimum gap between the print table and the roller surface) was set to 1.05 mm. All samples prepared with the Strangpresse Model 19 used the 3D printing parameters shown in Table 1 to fabricate the single strand and the single beads with and without roller compression during the printing process.

In addition, a parametric study was performed to understand the effect of three LAAM 3D printing process parameters on the micro void volume fraction and sphericity in the printed beads. Print parameters considered here include nozzle temperature, extrusion speed and nozzle height above the print bed. The full factorial parameter study includes twenty-seven single beads which were 3D printed and scanned using µCT. The 3D printing process parameters used in this study are shown in Table 2.

### 2.2. Image Acquisition Using µCT

Micro-computed tomography has attracted attention among non-destructive inspection techniques in various applications that seek to understand the three-dimensional characteristics of a material’s microstructure [18]. In polymer composites, µCT provides 3D microstructural information about the polymer matrix and particle reinforcement, including the shape, size, distribution, and orientation of fibers. In addition, µCT provides information related to the shape, size, and distribution of defects such as cracks and voids within the composite [19]. µCT systems require that the object or material of interest be placed in front of the X-ray generating source while rotating around its central axis. 2D images (projections) of the sample are captured at numerous fixed orientations as the sample is rotated one complete revolution. The collected data set is then reconstructed into 2D slices, which are processed to allow for visualization as a 3D rendering [24]. 

In this study, µCT scans were performed using the NSI X3000 µCT system (North Star Imaging, Rogers, MN, USA). One of the challenges presented when scanning composite materials at low resolution is that of supporting the sample since vibration during the scan may cause blurred images that reduces the accuracy of further analysis of the data set. To avoid sample vibration and wobbling during the scan, a suitable sample holder was designed, and 3D printed to secure SCF/ABS samples near the µCT detector. Before setting the scan parameters, it is necessary to understand the tradeoff between contrast and intensity while choosing the optimal scan voltage. Voltage should be set high enough to provide the needed intensity based on the density of the desired material. However, increasing the voltage causes a reduction in the contrast of the scanned images. Image contrast can also be enhanced by adjusting the X-ray beam current. In this study, the X-ray source with an acceleration voltage of 51 kV and a current of 150 µA was used for all scans, which provided adequate beam intensity and contrast in the images. Each sample was rotated 360 degrees at increments of 0.25 degrees during the scan, resulting in 1440 projections. The detector captured the transmitted X-ray beam signals and collected the 2D attenuation distribution image at each scan angle. All generated images had a resolution (pixel size) of 10 microns. 

The raw data generated from µCT were then reconstructed into virtual 2D slices using efX-CT software (North Star Imaging, Minnesota, USA). An outlier median filter was used to preserve the edges and remove noise, which provided clear images for further processing during the reconstruction step. The reconstructed data was then imported into VGStudio Max 3.4 (Volume Graphics GmbH, Heidelberg, Germany) for surface determination and micro void analysis. In this study, the VGDefX algorithm within the VGStudio Max porosity analysis module was used to evaluate the voxel data set for micro voids. The VGDefX algorithm evaluates each voxel to determine if it is part of a void based on its gray value [25]. In this study, the void max threshold was set to 28.78 for all porosity analyses based on the gray value information in the histogram of the scanned parts. Results of the porosity analysis provided information on each void as well as overall statistical information. 

## 3. Results and Discussion

### 3.1. Void Volume Fraction and Volume Fraction Distribution

Four 13 (wt%) SCF/ABS samples including a pellet, a single freely extruded strand, a single regularly deposited bead, and a single roller-compressed bead were each scanned as described above. Scanning was followed by a reconstruction step and micro void volume fraction measurement. Multiple views of the scans and void volume fraction along the coordinate directions of all four samples appear below (cf. Figure 2, Figure 3, Figure 4, Figure 5, Figure 6, Figure 7, Figure 8 and Figure 9) where nominal dimensions of all samples are provided in Table 3. Note that the direction of extrusion in all samples is identified as the Z-direction, and for bead samples that were printed onto the LAAM build platform, the Y-direction is defined normal to the print surface. 

Results from our SCF/ABS pellet appear in Figure 2 and Figure 3. As shown in Figure 2b voids within the SCF/ABS pellet were elongated in the Z-direction and were more rounded in the cross-X normal to the print direction as shown in Figure 2c. Moreover, Figure 2c illustrates that large voids were more centrally located, and voids appeared to have a higher concentration in the center of the pellet rather than near its edge. In comparison, Figure 3a,b shows that the volume fraction was more uniform across the width (X-direction) than through the thickness (Y-direction) and voids had a higher volume fraction in the center (having a maximum of 11.64%). The void volume fraction decreased along the length of the pellet (Z-direction) taking values from 9.04% to 5.78%, as shown in Figure 3c. Scan results showed that the overall average void volume fraction was 7.78% within the microstructure of the pellet indicating that micro voids were present in the raw material prior to LAAM processing. Therefore, it is clear that the printing process is not the only factor that causes micro void formation within the microstructure of the 3D-printed parts. Our results are supported by other researchers who observed a large number of voids in the cross-sectional area of carbon-fiber-reinforced ABS pellets using SEM techniques [26,27]. It is expected that the difference between the coefficient of thermal expansion of short carbon fibers and the ABS polymer matrix, in addition to moisture at the fiber-matrix interface, promote void formation within the pellet [6,28]. 

Figure 4 and Figure 5 consider the microstructure of a single strand freely extruded in air without deposition. Scan results indicate that the extrusion process promoted a significant increase in micro void volume fraction from 7.78% in the pellet to 17.2% (a 121.1% increase) in the single freely extruded strand. As shown in Figure 4b voids within the single strand were not elongated in the extrusion direction which was prominent in the pellet as shown above. However, the cross section in Figure 4c shows that voids were more rounded in the plane normal to the print direction compared to the pellet. Figure 4c also indicates there was more uniformity in void size azimuthally. In addition, voids were more randomly dispersed, oriented, and interconnected within the microstructure of the single freely extruded strand (cf. Figure 4b) compared to the pellet (cf. Figure 2b). Figure 5a,b show that void volume fraction along the X-direction and Y-direction showed similar behavior. The void volume fraction was highest near the center (20.82% along the X-direction and 20.65% along the Y-direction), and the lowest void volume fraction occurred at the edges (9.13% along the X-direction and 9.59% along the Y-direction). More uniformity in void size azimuthally can be seen in Figure 4c which is also confirmed in Figure 5a,b. Unlike the pellet microstructure, void volume fraction in the strand remained unchanged along the extrusion direction (Z-direction).

Figure 6 and Figure 7 consider beads extruded and deposited on the print platform; scan results show that the overall average micro void volume fraction reduced from 17.2% for the single freely extruded strand to 13.56% (a 21.2% reduction) within a regularly deposited single bead. Similar small scale FFF results were reported by Yang et al. [6] which showed that the void volume fraction was significantly lower within the microstructure of an FFF short carbon-fiber-T300-reinforced nylon-6 composite bead deposited on the print bed (0.67%) than the extruded strand in the air (3.59%). Figure 6b shows that voids were far less elongated in the extrusion direction compared to the pellet (cf. Figure 2b) and were more spherical in shape as in the single strand (cf. Figure 4b). Figure 6b,c show that voids were more interconnected with irregular shapes within the microstructure of the regularly deposited single bead than the pellet and single strand. Further, volume fraction measurements in Figure 7a across the width of the bead (X-direction) show that the void volume fraction was higher near the center of the bead (17.1%) compared to the outer edges of the bead (8.26%). Additionally, through the thickness of the bead (Y-direction), the void volume fraction was significantly lower near the print bed (4.21%) and near the top surface (5.79%) compared to the other regions as shown in Figure 7b. Sommacal et al. [10] evaluated the void volume fraction within the microstructure of the carbon-fiber-reinforced PEEK which showed similar results for FFF beads indicating a lower void volume fraction near the print bed. As in the single strand, the void volume fraction in the regularly printed bead remained uniform along the direction of extrusion (Z-direction) since the printing process was performed with constant parameters along the length of the bead, as shown in Figure 7c.

Figure 8 and Figure 9 include results from the roller-compressed bead sample. Here, µCT results show that applying a roller as part of the printing process reduced the overall void volume fraction from 13.56% in the regularly deposited bead to 10.12% in the roller-compressed bead (a 25.4% reduction). As shown in Figure 8b voids were not elongated in the extrusion direction as seen in the pellet (cf. Figure 2b) and did not exhibit a more spherical shape as seen in the single strand (cf, Figure 4b). Additionally, unlike the regularly printed bead, the roller-compressed bead void distribution was significantly lower in the center of the bead compared to the edges along the width of the bead (X-direction). A significant reduction in void distribution occurred in the center along the thickness of the bead (Y-direction) as shown in Figure 8c. In addition, Figure 9a illustrates that along the width of the roller-compressed bead, the void volume fraction was significantly higher near the edges (13.3%) compared to the center (8.04%). Based on these results, it is apparent that the roller reduces voids near the center while increasing those near the edges of the bead. Figure 9b indicates that the void volume fraction was lowest near the print bed along the thickness direction (Y-direction), and a significant void volume fraction reduction occurred through the center of the roller-compressed bead. Similar to the single strand and regularly deposited bead, the void volume fraction remained unchanged along the extrusion (Z-direction), as shown in Figure 9c.

### 3.2. Comparison of Micro Void Shape 

Micro voids within the SCF/ABS samples were further analyzed to investigate their shape within the voxel data set. In this study, void sphericity was used to assess the shape of the void which is defined as the ratio between the surface area of a sphere with the same volume as the void and the surface area of the detected void given as
(2)$$\mathrm{Sphericity}=\frac{A_{sphere}}{A_{void}}$$

A perfectly spherical void has the sphericity value of 1, and void sphericity decreases as voids take on more irregular shapes such as that which occurs with void elongation. To illustrate sphericity, sphericity values for voids containing 1, 2, 3, and 4 voxels and the schematic of the voxels’ arrangement appear in Table 4. In addition, several irregularly shaped voids taken from our SCF/ABS samples are shown in Table 5.

In our study, voids composed of less than five voxels were discarded in the sphericity calculations to minimize the influence of voxel size on void sphericity results. A two-parameter Weibull distribution, given as
(3)$$f\left(x\right)=(b/a){\left(x/a\right)}^{\left(b-1\right)}e^{-{\left(x/a\right)}^{b}}\;x\geq 0$$
was used to model the void sphericity distribution where *a* and *b* are the Weibull scale and shape parameter, respectively. A histogram of the measured data along with the Weibull curve fit showing the fraction of voids over the range of measured sphericity for all scanned samples appears in Figure 10. Data for each sample producing the fitted curves and Weibull parameters appear in Figure 11 and Table 5, respectively.

Void sphericity was measured for all detected voids over five voxels within the microstructure of four scanned parts. A histogram for each of our SCF/ABS samples appears in Figure 10 which shows that the fraction of voids over the range of measured sphericity had a negatively skewed distribution (cf. Table 6). Results show that the extrusion process had little effect on the void sphericity within the microstructure where the mean sphericity was 0.621 within the freely extruded strand compared to a mean sphericity of 0.611 within the pellet. The LAAM deposition/extrusion process that forms our regularly printed bead and that using the roller during the print process caused the formation of larger voids and interconnected voids within the microstructure of the bead, as shown in Figure 6c and Figure 7c. Prior research showed that void sphericity is inversely proportional to void sizes [29,30] in which we found a similar outcome for the majority of the detected voids. Results of our study indicated a mean void sphericity of 0.565 and 0.529 within the regularly printed bead and roller-compressed bead, respectively. Void sphericity mean, standard deviation, and coefficient of variation appear in Table 7.

### 3.3. Effect of Printing Process Parameters on Void Volume Fraction

Previous work by Somireddy et al. [19] indicated that the optimal printing speed, melt temperature and layer thickness can minimize the microstructural voids within the parts manufactured using FFF. To better understand the effect of the printing process on micro void formation in LAAM, a three-factor parametric study was performed where twenty-seven LAAM single beads were 3D printed at various values of nozzle temperature, extrusion speed, and nozzle height above the print bed. The roller was not used in this parameter study so that all results shown here are for regularly extruded/deposited beads. All 3D-printed beads were scanned using the same scan parameters and procedures described above, and micro void volume fraction and sphericity were calculated within the microstructure of each bead. The printing process parameters are given in Table 8.

The mass flow rate was measured for all reported extrusion speeds by weighing a printed bead sample produced with a specified print duration. The wall shear rate for each flow rate was then computed assuming capillary viscometric flow in the nozzle exit flow channel and applying the Rabinowitsch-corrected shear rate equation (see, e.g., Duty et al. [31]). A density of 1.154 g/cm^3^ and a power law index of *n* = 0.450 reported by Wang et al. [32] for the same material was used to calculate the wall shear rate. The mass flow rate and wall shear rate for all reported extrusion speeds are shown in Table 9.

Results appearing in Figure 12 show that increasing the extruder RPM (i.e., increasing extrusion flow rate and wall shear stress) from 70 rpm to 90 rpm and increasing the nozzle temperature from 200 °C to 220 °C caused a reduction in the void volume fraction within the microstructure of a single regularly deposited bead. This reduction was likely related to the increase in shear rate within the screw and print nozzle which reduced the molten polymer composite’s viscosity, possibly allowing voids to escape the molten polymer before the cooling process occurred. Results also revealed that decreasing the nozzle height above the table from 1.2 mm to 1.0 mm reduced the void volume fraction. This reduction was likely due to the increase in flow pressure at the nozzle tip caused by the back pressure from the polymer composite melt being squeezed between the extrusion nozzle and the print bed. Prior research by Percoco et al. [33], also indicated that the flow counterpressure reduces by increasing the layer height of the bead. Previous work by Ning et al. [34], presented the effects of FFF process parameters on the mechanical properties of SCF/ABS, which indicated an increase in tensile strength and Young’s modulus of the composite with increasing nozzle temperature from 200 °C to about 220 °C beyond which a reverse in behavior was observed due to an increase in micro voids within the microstructure. In our study, we observed a similar trend where the void volume fraction decreased with increasing nozzle temperature from 200 °C to about 220 °C.

Void sphericity was also measured for all detected voids within the microstructure of the twenty-seven scanned parts, and mean void sphericity was calculated. Results of the effect of printing process parameters on mean void sphericity appear in Figure 13. Overall, extrusion rate, deposition height, and temperature appear to had little effect on sphericity as all values in Figure 13 were within a relatively tight range of 0.53 to 0.58. Although no correlation was found between the mean void sphericity and changing the extrusion speed and nozzle temperature, results show that increasing the nozzle height above the table in some parts yielded an increase in the mean void sphericity within the microstructure. For example, for a nozzle temperature of 220 °C and extrusion speed of 80 rpm, voids within the microstructure of the regularly printed bead were more spherical in shape with a nozzle height of 1.2 mm compared to a nozzle height of 1.1 mm and 1 mm.

## 4. Conclusions

This paper presents research that employs high-resolution 3D µCT to study the microstructure of the SCF/ABS pellet and additively manufactured single strand and single beads. Overall, the void volume fraction within the SCF/ABS pellet was 7.78%, while the void volume fraction in the single strand air-extruded microstructure increased to 17.20% due to the extrusion process. This study also examined the effect of using a roller during the printing process on the void volume fraction within the microstructure of a single roller-compressed bead and compared it to a regular single bead. The results showed that the void volume fraction in the single roller-compressed bead was 10.12%, while the void volume fraction was 13.56% in a regular single bead printed with the same printing process parameters. Results indicate using roller compression during the printing process significantly decreased the void volume fraction. µCT results showed that the void’s sphericity varied within the microstructure of the scanned parts. Voids had the highest mean sphericity within the single freely extruded strand and lowest mean sphericity within the roller-compressed bead. Deposition on the print bed appeared to reduce the void’s mean sphericity as did the application of roller-compression during the printing process. This study also evaluated the effect of the printing process parameters, including nozzle temperature, extrusion speed and nozzle height above the print bed. Results showed that increasing the extrusion speed from 70 rpm to 90 rpm and increasing the nozzle temperature from 200 °C to 220 °C resulted in a reduction in the void volume fraction within the microstructure of a single regular bead. This reduction is expected to be related to the increase in shear rate within the screw and print nozzle which reduces the molten polymer’s viscosity, possibly allowing voids to escape the molten polymer before the cooling process occurs. Results also reveal that decreasing the nozzle height above the table from 1.2 mm to 1.0 mm reduces the void volume fraction. This reduction is likely due to the decrease in pressure drops at the nozzle tip caused by the back pressure from the polymer composite melt being squeezed between the extrusion nozzle and the print bed. Results of the effect of the printing process parameters on mean void sphericity explain that extrusion speed, deposition height, and temperature have little effect on sphericity in which void sphericity variation was relatively small over all twenty-seven scanned regularly printed beads. Although no correlation was found between the mean void sphericity and changing the extrusion speed and nozzle temperature, results showed that increasing the nozzle height above the table in some beads caused an increase in the mean void sphericity within the microstructure.

## Figures and Tables

**Figure 1 polymers-14-05107-f001:**
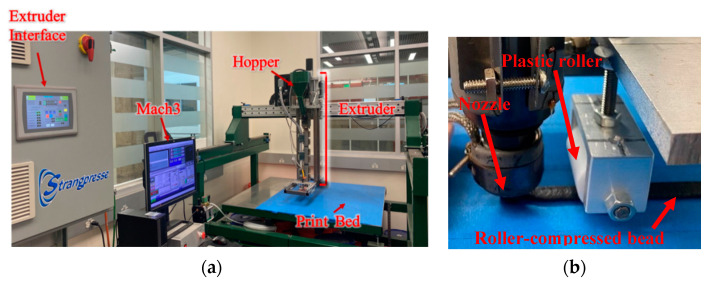
(**a**) Baylor’s custom large-area additive manufacturing (LAAM) system. (**b**) Plastic roller attached behind the nozzle.

**Figure 2 polymers-14-05107-f002:**
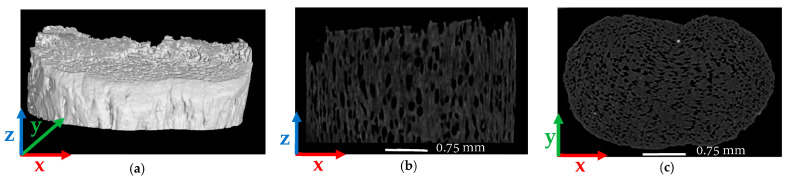
µCT images of SCF/ABS pellet: (**a**) 3D view; (**b**) 2D front view; (**c**) 2D top view.

**Figure 3 polymers-14-05107-f003:**
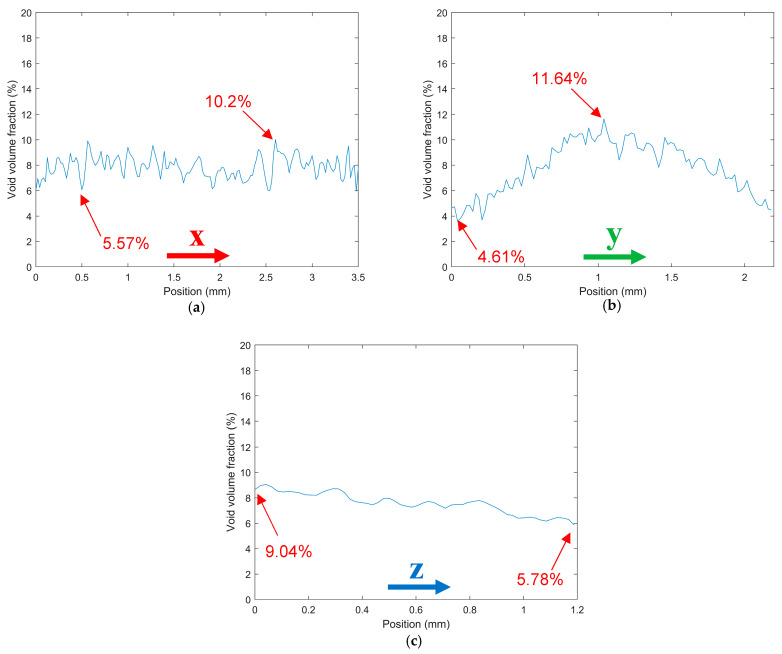
Void volume fraction within the different directions of a SCF/ABS pellet: (**a**) along the X-direction; (**b**) along the Y-direction; (**c**) along the Z-direction.

**Figure 4 polymers-14-05107-f004:**
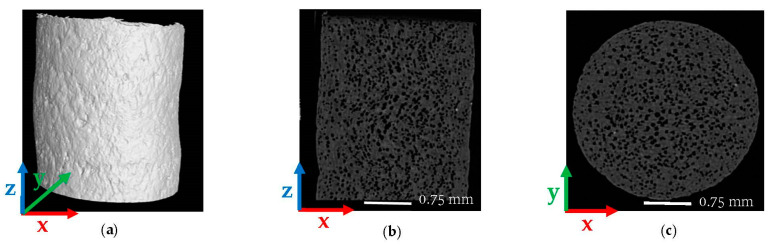
µCT images of a SCF/ABS single freely extruded strand: (**a**) 3D view; (**b**) 2D front view; (**c**) 2D top view.

**Figure 5 polymers-14-05107-f005:**
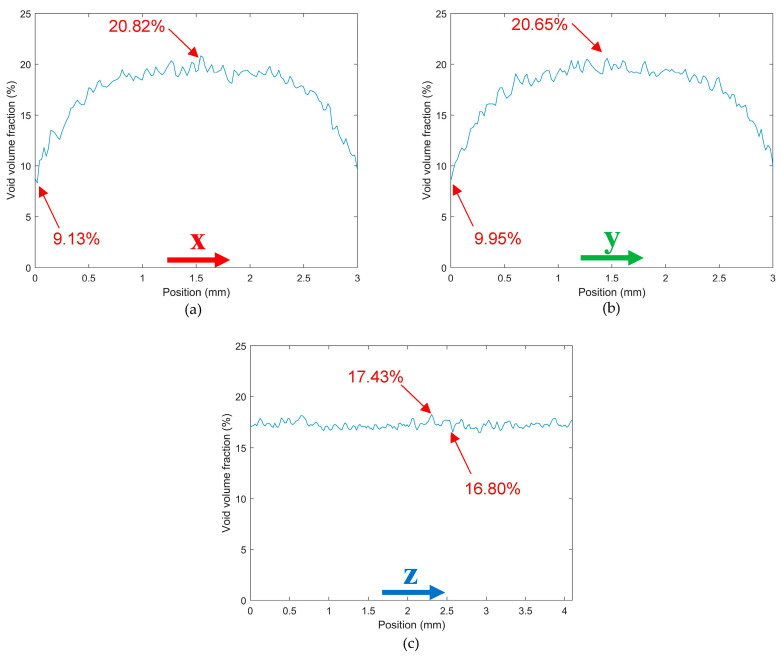
Void volume fraction along the coordinate directions of a SCF/ABS single freely extruded strand: (**a**) along the X-direction; (**b**) along the Y-direction; (**c**) along the Z-direction.

**Figure 6 polymers-14-05107-f006:**
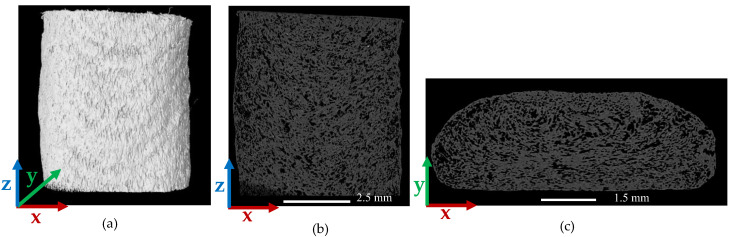
µCT images of a SCF/ABS regularly deposited single bead: (**a**) 3D view; (**b**) 2D front view; (**c**) 2D top view.

**Figure 7 polymers-14-05107-f007:**
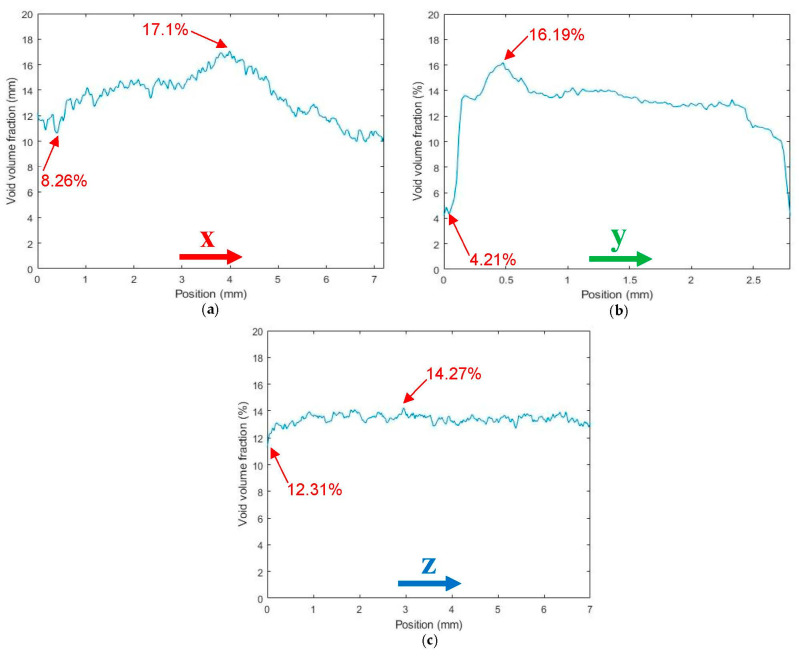
Void volume fraction along the coordinate directions of a SCF/ABS regularly deposited single bead: (**a**) along the X-direction; (**b**) along the Y-direction; (**c**) along the Z-direction.

**Figure 8 polymers-14-05107-f008:**
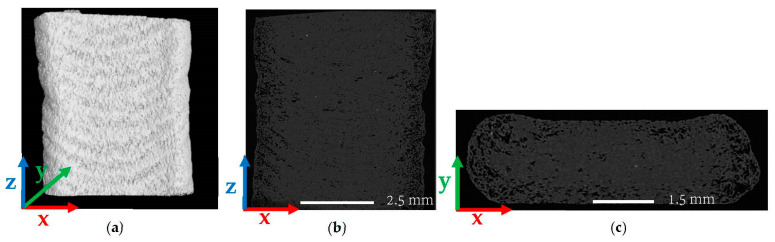
µCT images of a SCF/ABS roller-compressed bead: (**a**) 3D view; (**b**) 2D front view; (**c**) 2D.

**Figure 9 polymers-14-05107-f009:**
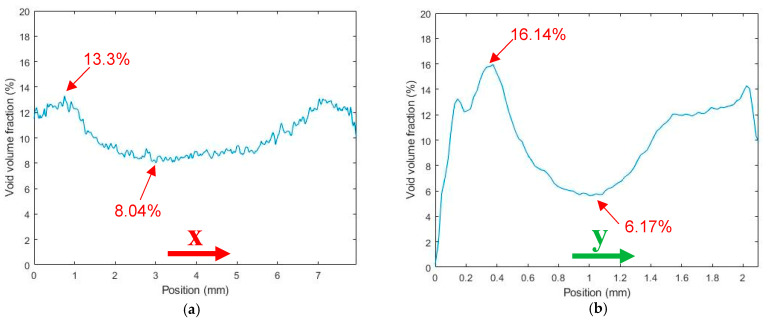
Void volume fraction along the coordinate directions of a SCF/ABS roller-compressed single bead: (**a**) along the X-direction; (**b**) along the Y-direction; (**c**) along the Z-direction.

**Figure 10 polymers-14-05107-f010:**
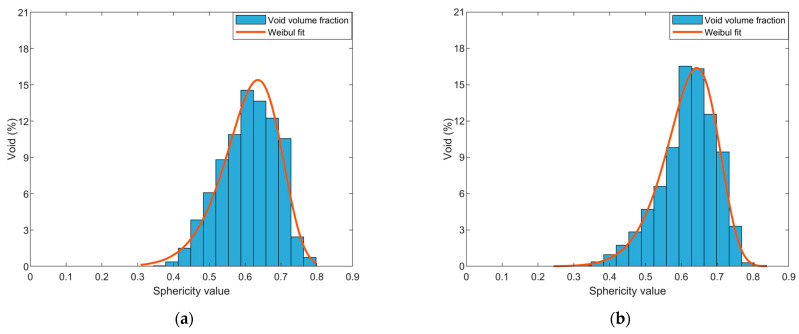
Fraction of voids sphericity histogram and Weibull fit of (**a**) a SCF/ABS pellet (**b**) extruded strand (**c**) regularly deposited bead, and (**d**) roller-compressed bead.

**Figure 11 polymers-14-05107-f011:**
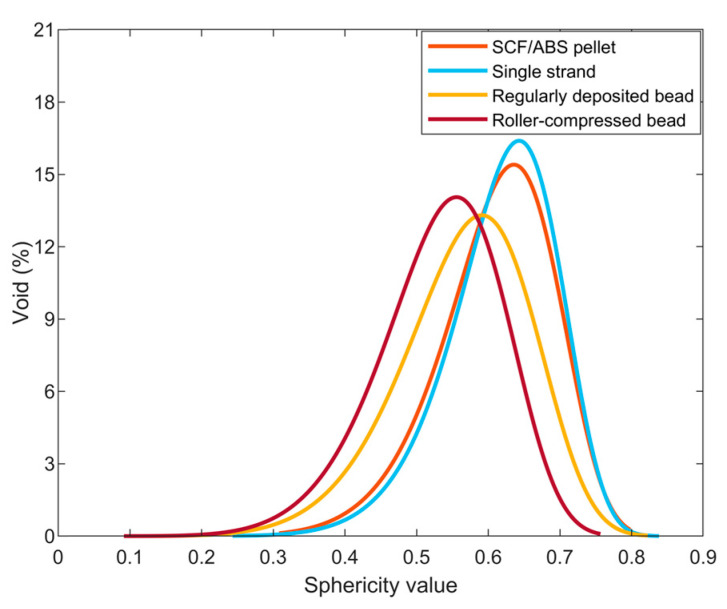
Weibull distribution of void sphericity in four different scanned parts.

**Figure 12 polymers-14-05107-f012:**
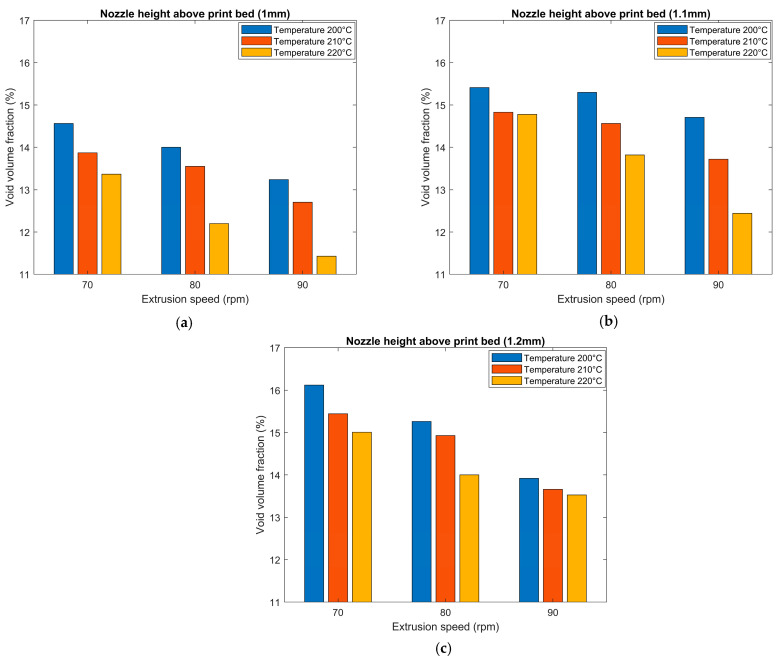
Effect of printing process parameters on void volume fraction for: (**a**) nozzle height (1 mm); (**b**) nozzle height (1.1 mm); (**c**) nozzle height (1.2 mm).

**Figure 13 polymers-14-05107-f013:**
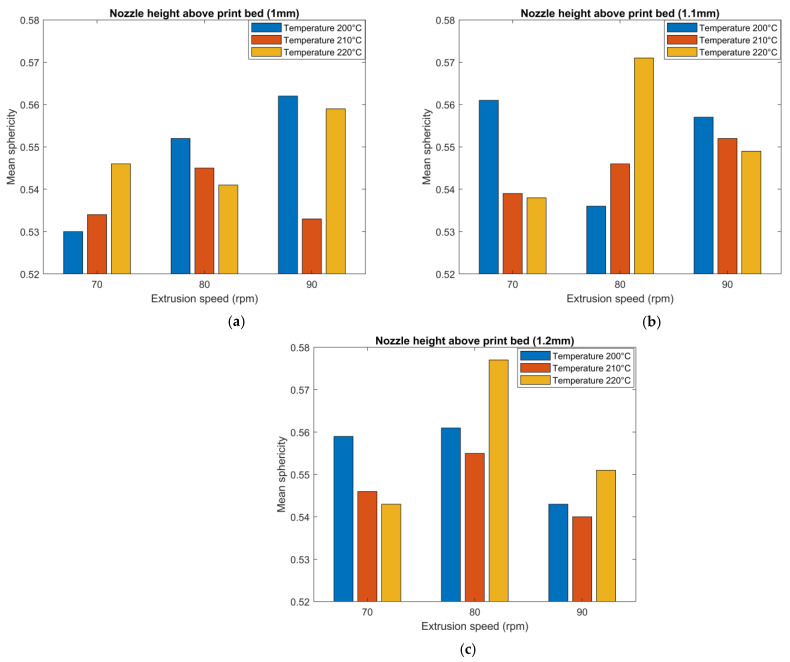
Effect of printing process parameters on mean void sphericity for: (**a**) nozzle height (1 mm); (**b**) nozzle height (1.1 mm); (**c**) nozzle height (1.2 mm).

**Table 1 polymers-14-05107-t001:** LAAM SCF/ABS extrusion/deposition printing parameters.

Printing Parameter	Value
Temperature	220–225–230 °C
Nozzle Height	1.20 mm
Roller Height	1.05 mm
Screw Speed	90 rpm
Nozzle Diameter	3.17 mm

**Table 2 polymers-14-05107-t002:** Printing parameters were used to 3D print twenty-seven single beads.

Temperature (°C)	Extrusion Speed (rpm)	Nozzle Height (mm)
200–205–210	70	1
210–215–220	80	1.1
220–225–230	90	1.2

**Table 3 polymers-14-05107-t003:** Nominal dimensions of SCF/ABS.

Sample	X-Direction (mm)	Y-Direction (mm)	Z-Direction (mm)
SCF/ABS pellet	3.5	2.5	1.2
Single strand	3.0	3.0	4.4
Regular bead	7.2	2.9	7.0
Roller-compressed bead	7.8	2.2	10

**Table 4 polymers-14-05107-t004:** Schematic of the voxels’ arrangement for several void sphericity.

Number of Voxels	Sphericity	Schematic of Voxels Arrangement
1	0.805	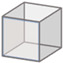
2	0.770	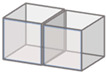
3	0.720	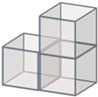
3	0.720	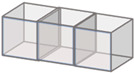
4	0.760	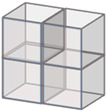
4	0.680	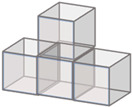

**Table 5 polymers-14-05107-t005:** Sphericity and number of voxels of several detected voids within the microstructure of a single freely strand.

Number of Voxels	Sphericity	Schematic of Voxels Arrangement
577	0.33	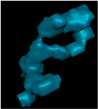
310	0.40	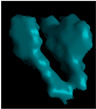
77	0.51	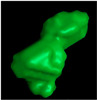
70	0.62	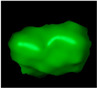
16	0.70	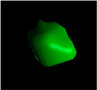
12	0.79	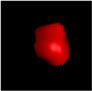

**Table 6 polymers-14-05107-t006:** Weibull distribution parameters of void sphericity in four scanned parts.

Part	Weibull Distribution Scale Parameter, a	Weibull Distribution Shape Parameter, b
SCF/ABS pellet	0.644	8.929
Single strand	0.650	9.603
Regular bead	0.603	7.197
Roller-compresses bead	0.567	7.150

**Table 7 polymers-14-05107-t007:** Mean, standard deviation, coefficient of variation and skewness of void sphericity in four scanned parts.

Sample	Mean	Standard Deviation	Coefficient of Variation	Skewness
SCF/ABS pellet	0.611	0.081	13.25%	−0.158
Single strand	0.621	0.078	12.56%	−0.353
Regular bead	0.565	0.097	17.16%	−0.454
Roller-compresses bead	0.529	0.094	17.76%	−0.604

**Table 8 polymers-14-05107-t008:** Printing process parameters.

Temperature (°C)	Extrusion Speed (rpm)	Nozzle Height (mm)
200	70	1
210	80	1.1
220	90	1.2

**Table 9 polymers-14-05107-t009:** Extrusion/deposition melt flow parameters for 200 °C nozzle temperature.

Extrusion Speed (rpm)	Mass Flow Rate (g/s)	Shear Rate (s^−1^)
70	0.80	288.4
80	0.91	329.3
90	1.04	397.7

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
