# Peer review of "Effect of Process Parameters on Void Distribution, Volume Fraction, and Sphericity within the Bead Microstructure of Large-Area Additive Manufacturing Polymer Composites"

_polymers, 2022, doi:10.3390/polym14235107_

Round 1

Reviewer 1 Report

Review report

Manuscript title: Effect of Process Parameters on Void Distribution, Volume  Fraction, and Sphericity Within the Bead Microstructure of  Large Area Additive Manufacturing Polymer Composites

Manuscript ID: polymers-2000761

Review comments: The work is very good and it has a novelty. The work demonstrates the effect of printing process parameters including, nozzle temperature, extrusion  speed and, nozzle height above the print bed on void volume fraction within the micro-  structure of SCF/ABS.  I think it will be published in the current journal after modification the issues raised. 

1.                  In figure 1 the image resolution is not good.

2.                  In table 3 the X,Y & Z-direction unit should be up-in one place, need not every row.

3.                  In figure 3, 5,7 and 9,  the x and y-axis text should be more clear.

4.                   In figure 12 and 13,  it needs high resolution.

5.                  In the discussion part need more references for comparison to others works.

6.                  There are many grammatical mistakes. The English needs to be rechecked and make it more corrections.

Author Response

The authors would like to thank the reviewers for providing the constructive feedback in the Reviewer Report below. In the following, we provide a detailed response (highlighted in yellow) to each of the reviewer comments. We believe these resulting manuscript edits and modifications have increased the value of our paper. Manuscript edits that address each of the comments are also highlighted in yellow in the resubmitted paper. We have chosen to identify changes to the manuscript in this manner rather than using ‘track changes’ since the latter would be quite cumbersome given that we have replaced most of the figures in the paper. We appreciate your input.

N. Sayah and D.E. Smith

Reviewer 2 Report

The manuscript entitled “Effect of Process Parameters on Void Distribution, Volume Fraction, and Sphericity Within the Bead Microstructure of Large Area Additive Manufacturing Polymer Composites” by Sayah and Smith investigates void volume fraction and distribution in pellets and printed beads (and strands) of short carbon fiber reinforced ABS samples using high resolution 3D micro-computed tomography technique. The authors report a marked increase in the void fraction during extrusion-based printing of the composite and employed a roller compression procedure to reduce the void fraction. Further, they studied the effects of print temperature, print speed, and nozzle height above the print bed on the void formation and its distribution in the printed part. However, the effect of the printing parameters on the sphericity of the voids was found to be negligible. The work performed is original and is of significant scientific interest to the additive manufacturing community. This reviewer suggests a few revisions, as outlined below, to the manuscript before its acceptance for publication.

Comments to the authors:

1.     The manuscript is heavily focused on AM; it would be nice if the authors could supplement some of their findings with polymer processing results (such as discussions regarding the materials’ rheology) to make it more interesting to the general audience of the journal.

2.     All the images except the micro-CT scans need to be significantly improved to meet the journal expectations (Please ensure all the graphs are at least 600 DPI)

3.     Figures 12 & 13: Please remove the solid lines joining the data points. Also, please add the standard deviation in these figures. The rotation speed can be replaced by calculating the associated shear rates.

4.     The authors should compare at least one of their results (void distribution/fraction) with the case of bench scale extrusion 3D printing (FFF). This will serve as a control against which the LAAM data is compared.

5.     Figure 5: The authors should comment on why the z-direction void volume fraction is relatively constant.

6.     The authors should comment on strategies to minimize the void fraction generation in the printed parts.

7.     The temperature distribution in the pellet, bead, and strand must either be measured or simulated to tie it with the void fraction results. Alternatively, they can refer back to previous works in the field:

a.     https://doi.org/10.1016/j.addma.2017.07.006

b.     https://doi.org/10.1016/j.addma.2020.101239

c.     https://doi.org/10.1016/j.addma.2022.102853

d.     https://doi.org/10.1016/j.addma.2019.04.014

8.     Discussions regarding how the void fraction, void fraction distribution, and sphericity affects the final printed part properties is missing. This reviewer will be highly interested in seeing this data incorporated into the manuscript along with the material rheology.

9.     Details regarding the sphericity calculation and a sample case can be included in the supplementary information.

10.  The fiber content and the fiber length distribution must be characterized post-printing and coupled with the results and discussions.

11.  This reviewer feels that having SEM of the fracture surfaces of the strands, beads, and pellets is necessary to understand the mechanism behind failure. If the prints fail due to poor interfacial adhesion between the fibers and the polymer matrix, then please comment on how important the relative effect of void formation and void distribution is.

12.  The Introduction is missing some relevant general literature sources; a non-exhaustive list of such publications is provided below:

a.     https://doi.org/10.1016/j.jmapro.2019.06.015

b.     https://doi.org/10.1016/j.addma.2019.03.030

c.     https://doi.org/10.1016/j.addma.2020.101218

d.     https://doi.org/10.1002/pc.26281

e.     https://doi.org/10.1016/j.addma.2020.101255

f.      https://doi.org/10.1108/RPJ-05-2019-0142

g.     https://doi.org/10.1122/8.0000052

Author Response

(The authors gave the same response as above.)

Round 2

Reviewer 2 Report

Thank you for submitting the revised manuscript incorporating the reviewers' comments. The quality of the manuscript has improved significantly.